# An image processing technique for optimizing industrial defect detection using dehazing algorithms

**Xuanyi Zhao**◉, **Xiaohan Dou**◉, **Gengpei Zhang**◉*

Yangtze University, Jingzhou, Hubei, China

◉ These authors contributed equally to this work.
* judgebill@126.com

## Abstract

In recent years, the demand for efficient and accurate defect detection algorithms in industrial production has been increasing. However, industrial cameras may be affected by water fog during image acquisition, resulting in image blurring and quality degradation, which increases the difficulty of defect detection. This paper proposes an industrial defect detection algorithm incorporating dehazing technology to enhance detection performance in complex environments. Experimental results show that using an optimized dehazing processing method on industrial images affected by water fog achieves an average PSNR of 34.9 dB and an SSIM of 0.951. The overall performance surpasses CNN and MADNet models, and verification using the improved YOLOv8 model significantly enhances defect detection confidence while greatly reducing missed detections. Further research indicates that this method is not only applicable to industrial defect detection but can also be transferred to personnel localization and rescue tasks in fire and smoke environments. This study provides a novel technical approach for industrial defect detection in complex environments and offers valuable references for image processing and object detection tasks in other fields.

## 1. Introduction

The development of Industry 4.0 has led to the widespread application of automated visual inspection systems in quality control, fault diagnosis, and production line monitoring, becoming a key technology for improving production efficiency and ensuring product quality. However, industrial environments are often accompanied by various adverse factors that severely impact image quality, consequently affecting the performance of defect detection systems. Common factors causing decreased accuracy in industrial vision systems include haze, dust, moisture, low-light conditions, and blurring due to long-distance shooting. This problem is particularly prominent in special environments such as chemical and petroleum industries.

**Data availability statement:** Only part of the data was used in the study, and it was guaranteed that all relevant data used were included in the manuscript and its supporting information files.

**Funding:** The author(s) received no specific funding for this work.

**Competing interests:** The authors have declared that no competing interests exist.

Chemical plants are often exposed to high-temperature, high-pressure, and highly corrosive chemical substances, generating large amounts of steam and gas during production. These airborne gases cause a haze effect in the imaging environment of visual systems, significantly reducing image contrast and detail, thereby affecting the accuracy of defect detection algorithms. In chemical production, defects such as cracks, corrosion, and leaks are likely to occur in pipe connections, valves, and equipment surfaces. Traditional manual inspection struggles to meet the demands of high-risk environments, whereas machine vision-based automated systems can provide more efficient and precise inspection solutions. However, the presence of haze makes it difficult to clearly identify surface defects on these devices, affecting the reliability of detection systems.

In the petroleum industry, particularly in deep-sea and onshore oilfield exploration and production, the environmental conditions are even more complex. During the monitoring of equipment such as oil wellheads and drilling platforms, image blurring often occurs due to oil and gas volatilization, dust, and temperature fluctuations, making it difficult to distinguish target objects due to low contrast or blurring. Defects such as minor cracks, corrosion, and leaks in pipelines, oil transportation equipment, and oil storage tanks must be detected promptly to prevent severe safety accidents or production interruptions. Therefore, the petroleum industry urgently requires efficient visual inspection technologies to ensure the safety and stable operation of equipment, as shown in Fig 1.

Given the need for visual inspection in such complex environments, the application of dehazing technology is particularly important.

In recent years, significant progress has been made in image dehazing research. With the diversification of application scenarios and increasing demands for real-time processing, physical dehazing methods have gained attention. These methods directly intervene in the imaging process through hardware means to reduce the impact of fog on image quality. For example, the application of anti-fog films or coatings alters the physical properties of lens surfaces to reduce water vapor condensation, thereby preventing lens fogging effectively [1]. Additionally, integrating automatic wiping devices enables rapid removal of water vapor from lens surfaces when fog is detected, which is particularly useful in real-time scenarios such as autonomous driving and surveillance systems [2]. The introduction of heating elements can also disperse fog on lens surfaces, significantly improving image clarity in low-temperature and high-humidity environments [3].

Although hardware-based dehazing techniques are effective in certain scenarios, they have notable disadvantages. Hardware dehazing increases system costs and complexity due to the need for additional components such as anti-fog films, automatic wiping devices, or heating elements. The maintenance and replacement of these hardware components also add to operational costs. Additionally, the applicability of hardware dehazing is limited to specific environmental conditions. For instance, anti-fog films and heating elements perform well under controlled temperature and humidity conditions but may not function effectively in extreme weather (e.g., freezing temperatures or high humidity). While automatic wiping devices can

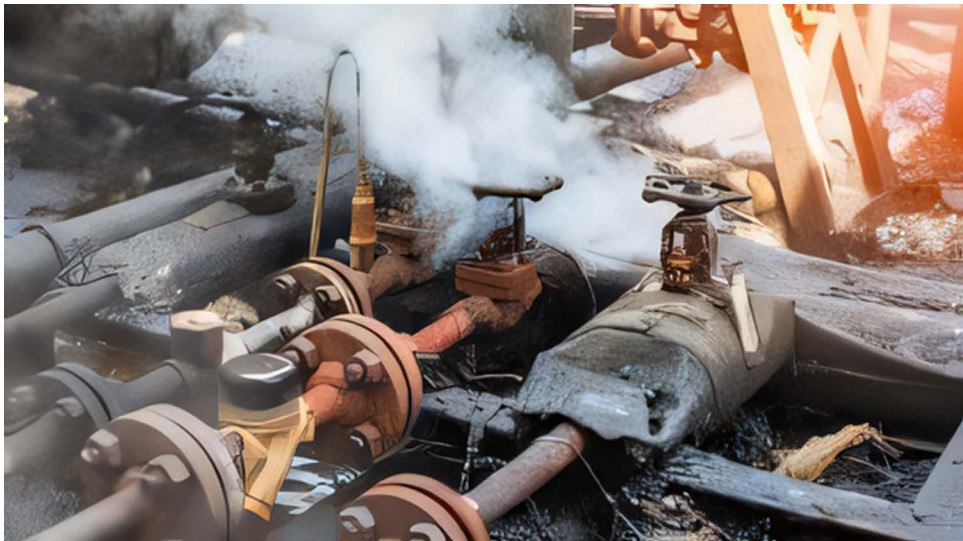

**Fig 1. Industrial smoke vapor.**

remove fog from lens surfaces in real time, they cannot handle persistent or rapidly changing fog conditions, and frequent wiping may damage the equipment or degrade image quality. Furthermore, hardware solutions respond slowly, may fail to adapt to sudden environmental changes, and typically only remove fog from lens surfaces without addressing optical losses and image detail blurring caused by haze.

In contrast, algorithm-based dehazing offers significant advantages. It provides greater flexibility and adaptability, as image processing techniques can dynamically adjust parameters to optimize dehazing effects without relying on specific environmental conditions. This makes algorithmic dehazing highly suitable for real-time processing applications, requiring no physical intervention and allowing immediate processing of images after acquisition without specialized hardware support.

Dehazing algorithms restore light scattering effects in images, enhancing visibility and contrast to improve overall image quality. Meanwhile, deblurring techniques restore blurred details, making object contours clearer, thus providing more precise image data for subsequent defect detection. The combined use of these methods effectively overcomes image quality issues caused by environmental factors, offering clearer and more reliable data for industrial inspection. To address these challenges, Kim et al. [4] proposed an optimized contrast enhancement technique for real-time image and video dehazing, focusing on improving image quality by enhancing contrast. Li et al. [5] comprehensively evaluated existing single-image dehazing algorithms using large-scale benchmarks, including both synthetic and real-world foggy images. They also proposed various evaluation criteria for dehazing algorithms, including full-reference metrics, no-reference metrics, and subjective evaluation. In a recent study, Li et al. [6] introduced an effective data-driven single-image dehazing method to enhance the efficiency of outdoor vision systems. Compared with existing methods, this approach demonstrated superior stability and dehazing performance. Jia et al. [7] developed a meta-attention dehazing network (MADN) tailored to the specific requirements of industrial systems. This network is designed to directly restore clear images from foggy ones without relying on physical scattering models. Wu et al. [8] proposed a contrastive regularization technique for compact single-image dehazing, utilizing contrastive learning to enhance the restoration process. Ullah et al. [9] introduced Light-DehazeNet, a lightweight CNN architecture for single-image dehazing that jointly estimates transmission maps and atmospheric light using a transformed atmospheric scattering model. Song et al. [10] developed WSAMF-Net, a wavelet spatial attention-based multi-stream feedback network for single-image dehazing. Additionally, Liu et al. [11] proposed a

novel objective evaluation metric, SAGE-NDVI, for remote sensing image dehazing assessment, breaking stereotypes in dehazing evaluation methods. Zhuang et al. [12] introduced the Dimensional Transformation Mixer (DMixer) model for ultra-high-definition industrial camera dehazing, enabling real-time processing of UHD images. Cui et al. [13] proposed LoGoNet, a local-global representation learning framework for image restoration, utilizing transformer-based architectures to improve performance.

This paper compares and selects the optimal image processing methods based on the aforementioned approaches to address defect detection challenges in specialized scenarios.

## 2. System theoretical foundation

The purpose of this paper is to develop an efficient industrial defect detection system. Firstly, the collected fuzzy image system is denoised, and advanced denoising algorithms are used to reduce sensor noise and environmental interference and improve image quality. Then, dehazing and deblurring are performed, and illumination enhancement is performed in special scenes to improve the clarity of the image and ensure that the defects can be obvious. After image preprocessing, the defect detection algorithm is applied to accurately identify and classify industrial defects. The proposed method provides reliable technical support for defect detection in complex industrial environments by optimizing image quality, including steps such as denoising, dehazing, and illumination enhancement, as shown in Fig 2.

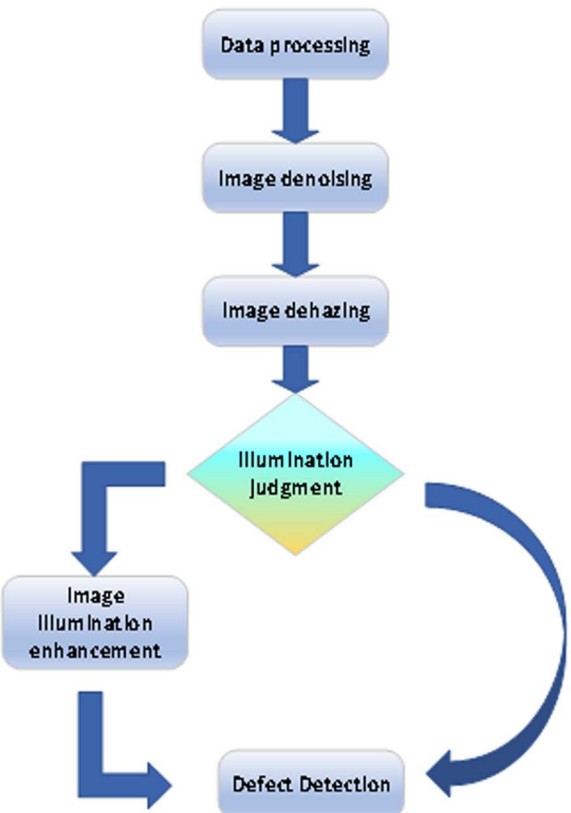

**Fig 2. Overall system framework.**

## 2.1 Noise processing

Industrial noise reduction faces major challenges due to the complex and diverse types of noise, including Gaussian noise, salt-and-pepper noise, Poisson noise, and other mixed types of noise. Additionally, there are significant differences in noise intensity under different devices and shooting conditions. In the denoising process, it is difficult to preserve image details, edge information is easily lost, and texture information is often damaged, yet these details are critical for industrial image recognition and measurement tasks. Meanwhile, industrial scenarios demand high real-time performance for image denoising, as the image data is large and computational resources are limited, making traditional denoising algorithms inadequate. Moreover, the difficulty in obtaining training data, with limited sample sizes and imbalanced data, affects the training performance and model generalization ability of deep learning-based denoising methods. Some denoising models overly depend on specific datasets, making them sensitive to noise variations, and they are hard to adapt to the changing noise characteristics in industrial scenarios due to factors such as time and environment, requiring constant retraining or parameter adjustments.

Zauner et al. [14] proposed a multi-scale segmentation method for detecting small inclusions in three-dimensional industrial CT scans. This method is based on multi-resolution denoising technology and aims to achieve high-precision detection of inclusions in cast iron samples. Dan et al. [15] introduced a wavelet image denoising algorithm with a directional window for local adaptive Wiener filtering, demonstrating its effectiveness in blast furnace image detection. Song et al. [16] focused on industrial image denoising using multi-wavelet Riesz bases and provided representations for forming two Riesz bases based on the tight frame equation. Komoto et al., 2018 [17] discussed the application of denoising autoencoders with generative adversarial networks (GANs) in industrial product defect detection, aiming to restore defect images to clearer defect-free images. Lee et al. and Kim et al. [18] both explored the application of non-local means denoising algorithms in non-destructive high-energy industrial X-ray imaging systems, with Kim et al. [19] particularly focusing on a 3 MeV linear accelerator high-energy X-ray system. Li et al. [20] proposed a lightweight GAN-based defect segmentation system for industrial IoT applications, emphasizing real-time production needs. Furthermore, Zhu et al. [21] introduced a GAN-based zero-shot learning method for denoising industrial positron emission tomography (PET) images, addressing the need for improved denoising performance in domains with limited sample data. Jiang et al. [22] developed an adaptive graph prior network (AGP-Net) for industrial image denoising applications, capturing long-range dependencies at the pixel and patch levels. Finally, Zhang et al. [23] proposed an improved KBNet for industrial digital radiography (DR) image denoising, enhancing the quality of industrial DR images affected by noise through lightweight modifications. These studies collectively emphasize the importance of customizing denoising methods to meet industrial image processing needs and demonstrate various methods to tackle noise-related challenges in industrial environments.

This paper addresses the diverse and complex noise types in industrial scenes by using Swin-Conv-UNet (SCUNet) to remove noise from images without prior knowledge of the noise type or noise level (such as Gaussian noise, Poisson noise, speckle noise, compression noise, and sensor noise, etc.).

SCUNet effectively captures the features of different noise types by combining local modeling (residual convolutions) and non-local modeling (Swin Transformer), thereby performing excellently in complex noise environments. Its design enables the model to handle various noise patterns, including those common in low-light and high-dynamic-range industrial applications. The network architecture uses a multi-scale U-Net backbone, which is particularly important for complex noise scenes in industrial images that need to process both fine details and overall structure simultaneously. By introducing the innovative Swin-Conv block, SCUNet enhances denoising performance by balancing local details and global structure, especially when processing industrial images with periodic patterns or repetitive structures. The model's non-local modeling ability further strengthens its performance in removing noise from images and restoring important details.

For blind image denoising, SCUNet models the process using the following Maximum A Posteriori (MAP) formula:

$$\hat{x} = \arg\min_{x} D(x, y) + \lambda P(x)$$

(1)

To tackle the diverse industrial noise, SCUNet employs a carefully designed training data synthesis pipeline that simulates combinations of multiple noise types. The pipeline expands the adaptability of the noise model through dual degradation strategies and random shuffling strategies. This design allows SCUNet to handle the complex noise patterns found in real industrial environments, thereby improving its performance and generalization ability in practical applications. Compared with traditional image denoising methods, SCUNet exhibits superior denoising performance under various industrial noise types. As shown in Fig 3, the data synthesis pipeline used by SCUNet demonstrates how high-quality images are converted into noisy images through a series of noise addition processes.

Additionally, SCUNet performs excellently in computational efficiency, reducing both computational complexity and parameter count while maintaining high denoising performance. Its optimized architecture enables SCUNet to run on edge devices or embedded systems, making it suitable for real-time industrial monitoring and automation systems. This means SCUNet not only delivers superior denoising performance but also can be widely applied in real industrial environments to meet the demands for real-time processing and efficiency.

## 2.2 Dehazing

In industrial scenes, image blurring caused by water fog or other environmental factors reduces image contrast and results in detail loss, which presents challenges for subsequent defect detection and quality control tasks. This paper selects CasDyF-Net to process industrial images captured in special environments, performing dehazing to improve image clarity. This method incorporates a multi-branch network, residual multi-scale modules (RMB), and dynamic fusion strategies, effectively retaining image details while enhancing the model's adaptability and computational efficiency. Fig 4 illustrates the overall structure and key modules of the proposed dehazing method, including dynamic filtering, local feature fusion (LFB), and global fusion mechanisms.

In this method, the dynamic feature segmentation mechanism generates convolution kernels through dynamic filters to extract features of different frequencies. Specifically, the dynamic filter dynamically generates convolution kernels based on the distribution of input features, and the process is as follows:

$$K_i = S\left(BN\left(W_i\left(GAP\left(X_i\right)\right)\right)\right), \quad i = 1, 2, \ldots, n-1 \tag{2}$$

Where $GAP$ represents global average pooling, $W_i$ is the convolution layer parameter, $BN$ represents batch normalization, and $K_i$ represents the dynamically generated convolution kernel. Then, the filtered feature branches are extracted through the convolution operation.

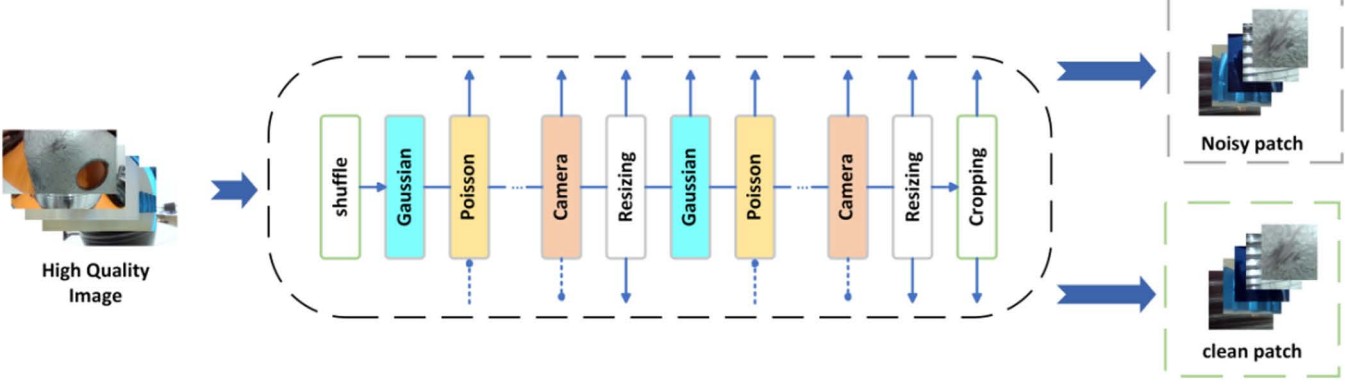

**Fig 3. SCUNet structured flowchart.**

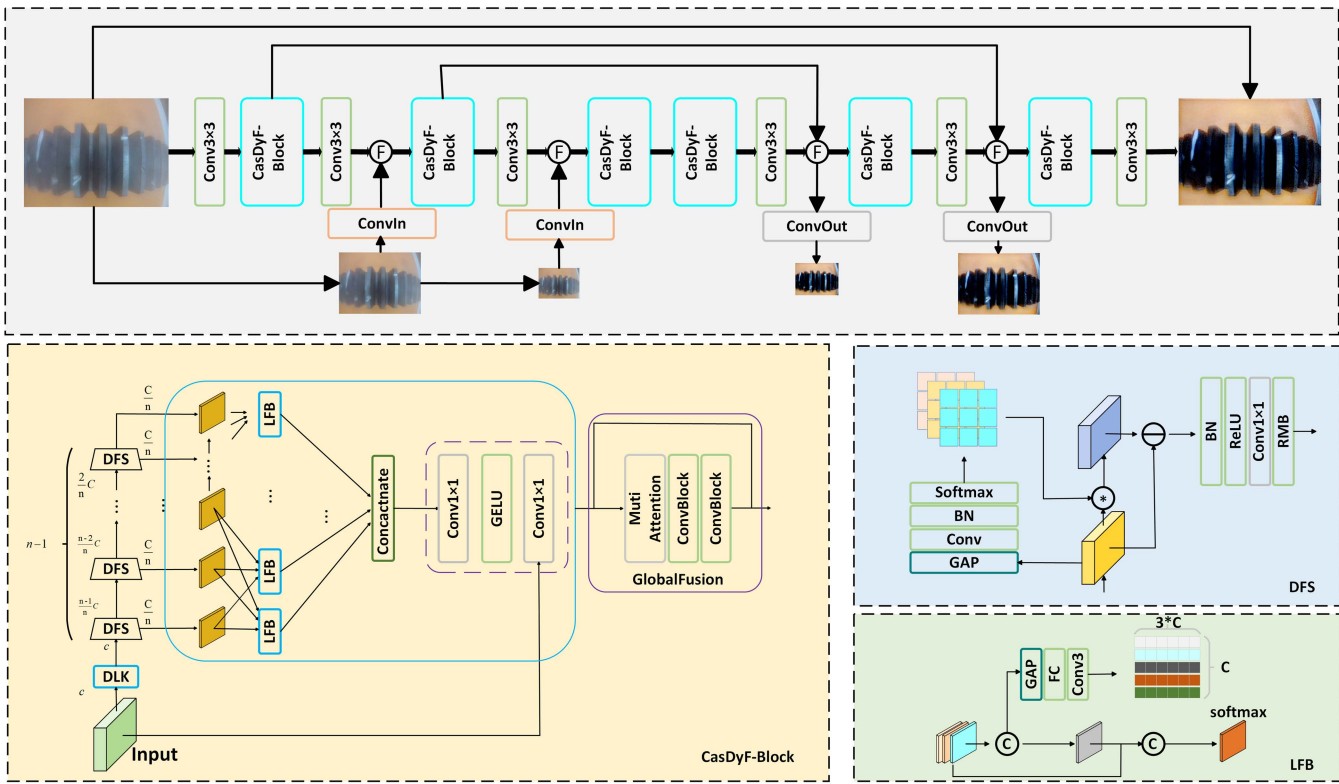

**Fig 4. SCUNet Network structure diagram.**

To further extract multi-scale information, this method introduces the Residual Multi-Scale Module (RMB). RMB uses convolution kernels with different dilation rates to capture multi-scale features of the image, thereby improving the representation of high-frequency details. When the industrial camera captures images blurred by water mist, this method can perform efficient image dehazing by dynamically fusing local and global features, restoring clear textures and details. Its feature extraction process is represented as:

$$F_i = RMB_i\left(W_i^{out}\left(ReLU\left(BN\left(Y_i\right)\right)\right)\right), \quad i = 1, \ldots, n-1 \tag{3}$$

Additionally, this method integrates multi-branch features through a dynamic fusion strategy to achieve efficient fusion of local and global information. In the local fusion stage, each branch dynamically fuses features with its neighboring branches, and the process is as follows:

$$F_i^l = \begin{cases} LFB_i\left(F_i, F_{i+1}, F_{i+2}\right), & i = 1 \\ LFB_i\left(F_{i-1}, F_i, F_{i+1}\right), & i = 2, \ldots, n-1 \\ LFB_i\left(F_{i-2}, F_{i-1}, F_i\right), & i = n \end{cases} \tag{4}$$

In the global fusion stage, a parallel attention mechanism is used to further integrate all branch features, ultimately generating a natural and clear dehazed image.

To ensure the dehazing effect, the method considers both spatial and frequency domain information in the loss function. The loss function is composed of the cumulative multi-scale loss, where each scale's loss contains both spatial domain loss and frequency domain loss:

$$L = \sum_{s=1}^{3} \frac{1}{E_s} \left( \parallel \hat{X}_s - X_s \parallel_1 + \lambda \parallel F\left(\hat{X}_s\right) - F(X_s) \parallel_1 \right)$$

(5)

Where $\hat{X}_s$ represents the model output, $X_s$ is the corresponding ground truth image, and $F$ represents the fast Fourier transform, $\lambda = 0.1$.

Through the above method, the dynamic filter can adaptively adjust based on the distribution of input features, solving the problem that fixed filters struggle to handle complex scenes. At the same time, the RMB module effectively integrates different frequency information through multi-scale convolutions, enhancing the restoration of high-frequency textures.

## 2.3 Illumination enhancement

In industrial image processing, water fog significantly impacts image quality, especially in low-illumination environments where water fog reduces contrast and causes detail blurring, thus affecting defect detection and object recognition tasks. While dehazing technology can effectively remove water fog, the resulting image may still suffer from low brightness or a hazy appearance, particularly in complex scenes and low-light conditions. Illumination enhancement can improve dark areas and low-contrast regions, restoring image details and textures, which is particularly important for industrial defect detection, as small defects often hide in low-contrast regions. Therefore, performing illumination enhancement after dehazing not only helps improve visual quality but also provides clearer visual information for subsequent image processing tasks. Enhanced images can provide clearer object boundaries and feature information in tasks such as image segmentation and object detection, thereby improving algorithm accuracy and robustness.

Light adjustment for industrial image enhancement has become a research hotspot in recent years. Li et al. [24] focused on contrast enhancement for raised character images, highlighting the difficulties in recognizing raised characters due to lighting factors. Similarly, Zafar et al. [25] proposed an image enhancement algorithm for industrial barcode reading, including non-uniform lighting correction. Wang et al. [26] proposed a naturalness-preserving non-uniform lighting image enhancement algorithm, emphasizing the importance of preserving natural details during the enhancement process. In the field of low-light image enhancement, Wei et al. [27] developed a deep Retinex network for light adjustment, denoising, and reflection enhancement. Kim et al. [28] proposed a dehazing and low-light enhancement method based on convolutional neural networks for precise light map estimation. Goyal et al. [29] addressed the challenge of balancing light intensity, detail representation, and color integrity in low-light image enhancement. Additionally, research has been conducted on industrial vision detection systems under complex lighting conditions. Liu et al. [30] proposed a method combining image enhancement, edge detection, and feature extraction for object detection in industrial robotic systems. Furthermore, Zou et al. [31] proposed a method for tile surface defect visual detection based on image enhancement and region-growing algorithms. In summary, recent research on industrial image illumination enhancement has garnered widespread attention, particularly in contrast enhancement, non-uniform lighting correction, low-light image enhancement, and visual detection under complex lighting conditions. These studies highlight the importance of preserving natural details, balancing enhancement characteristics, and using advanced algorithms for effective light adjustment in industrial applications.

This paper selects an illumination enhancement method that avoids the common pitfalls of existing LLIE methods, which typically rely on deep neural networks to learn mappings between low-light images and normal-light images in sRGB and HSV color spaces. However, these methods often suffer from instability and artifacts in the enhancement results due to sensitivity to color space variations. To address this issue, a new color space called Horizontal/Vertical Intensity (HVI) is introduced, which decouples brightness and color information to reduce instability during the enhancement process and to better accommodate low-light images with different illumination ranges.

The HVI color space introduces three trainable parameters and a custom training function that dynamically adjusts brightness and color variation in low-light images during enhancement. This method not only addresses the shortcomings of traditional color spaces (such as sRGB and HSV) in low-light image enhancement but also provides higher stability and accuracy in the process.

Building upon this, the paper introduces a low-light image enhancement method called Color and Intensity Decoupling Network (CIDNet). CIDNet uses a dual-branch structure, with the HVI color branch and intensity branch processing color and brightness information separately, and combines them through a Lighten Cross-Attention (LCA) module to promote interaction between the two branches. The LCA module uses a cross-attention mechanism to strengthen the complementarity of brightness and color information, effectively suppressing noise in low-light images while enhancing detail representation.

In the LLIE task, one key step is accurately estimating the scene's brightness intensity map. According to the existing Retinex theory, the intensity map is usually calculated by taking the maximum value of the three channels of the RGB image, as shown in equation (6):

$$I_{\max} = \max_{c \in \{R,G,B\}} (I_c)$$

(6)

This equation gives the maximum brightness value for each pixel and constructs the intensity map of the image.

To address the color loss in low-intensity areas of low-light images, the HVI color space introduces a trainable parameter $k$ to adjust the color point density in the low-intensity color map. This adjustment is described by the following equation (7):

$$C_k = k\sqrt{\sin\left(\frac{\pi I_{\max}}{2}\right) + \in}$$

(7)

where $k$ is a positive adjustment factor and $\in$ is a constant to prevent division by zero, ensuring that color details are maintained in the low-intensity regions.

To reduce color shifts caused by different camera sensors, a linear color perceptual mapping function $P_\gamma$ is designed to adaptively adjust based on the hue value. The chromatic adjustment process is shown in equation (8):

$$P_\gamma = \begin{cases} 3\gamma_G H, & \text{if } 0 \leq H < \frac{1}{3} \\ 3\left(\gamma_B - \gamma_G\right)\left(H - \frac{1}{3}\right) + \gamma_G, & \text{if } \frac{1}{3} \leq H < \frac{2}{3} \\ 3\left(1 - \gamma_B\right)\left(H - 1\right) + 1, & \text{if } \frac{2}{3} \leq H < 1 \end{cases}$$

(8)

This equation allows for effective adjustment of color shifts under varying lighting conditions, ensuring that the colors in the image remain natural.

The HVI color space is converted back to the sRGB space using the following inverse transformation, described in equation (9), ensuring that the enhanced image is correctly mapped back to the standard RGB space:

$$H = F_\gamma\left(\arctan\left(\frac{\hat{v}}{\hat{h}}\right) \bmod 1\right)$$

(9)

where $F_\gamma$ is an inverse linear function, restoring the original RGB image based on chromaticity and intensity information.

## 2.4 Defect detection in images

In this study, YOLOv8 [32] was selected as the base model for object detection due to its outstanding performance in real-time tasks, providing high computational efficiency while ensuring high accuracy. Compared to previous versions,

YOLOv8 has made improvements in feature extraction and computational graph optimization, making it particularly suitable for industrial defect detection. It can be efficiently deployed in various industrial scenarios, offering high-precision detection results and ensuring fast inference, making it an ideal choice for real-time applications in industrial defect detection. Its architecture achieves real-time deployment on edge devices through a lightweight design (with approximately 7.2M parameters and an inference speed of 0.8ms/frame on an RTX 3080), reducing memory usage by 32% compared to newer models like YOLOv12. Furthermore, the detection head module optimized for industrial defect characteristics (adaptive spatial feature fusion mechanism) maintains stable accuracy on standard datasets such as PCB (98.2% mAP) and metal surfaces (96.5% mAP). Additionally, its comprehensive toolchain supports ONNX/TensorRT conversion, seamlessly integrating with industrial vision systems and reducing deployment costs by 40%.

Although newer algorithms may achieve a 0.5–2% improvement in AP for specific scenarios (e.g., small-sample defect detection), the increased 15–30% computational complexity often exceeds the real-time threshold of industrial hardware, such as the Jetson AGX Xavier. Therefore, the primary reason for prioritizing YOLOv8 in industrial defect detection tasks is its triple advantage in stability, computational efficiency, and task optimization.

For the processed industrial images, the YOLOv8 model is employed for defect detection. The network structure is modified and enhanced for industrial part defect detection tasks. The overall network structure is shown in Fig 5.

In the Backbone, the AgentAttention module is added, which combines Softmax attention and linear attention. The core idea is to introduce a set of Agent tokens A into the traditional tuple-based attention mechanism (Q, K, V). These Agent tokens serve as proxies for Q, aggregating information from K and V, and feeding the processed information back to Q. Since the number of Q tokens is much smaller than the number of Agent tokens A, this significantly improves the efficiency of the Agent attention while retaining the ability to capture global context feature information. Additionally, Agent Bias and

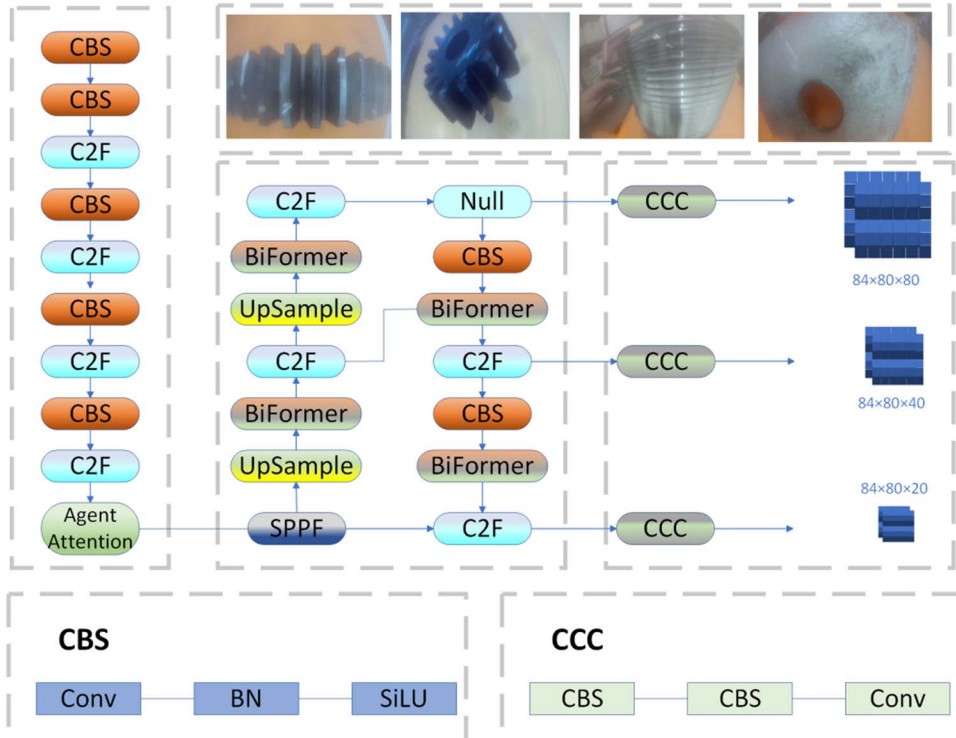

**Fig 5. YOLOv8 improved network structure diagram.**

DWC components are used to record positional information across different scales and feature maps, ensuring the accuracy and diversity of feature information. This enhances the model's ability to extract features from images in industrial scenarios, significantly improving the overall performance of deep learning models in defect detection under misty conditions.

The BiFormer module is adopted, which is based on a dual-level routing attention mechanism. By dynamically allocating resources, it achieves sparse sampling and adaptively fuses global and local features, enhancing the model's ability to detect targets at multiple scales and angles. Specifically, the BiFormer module uses a four-stage pyramid structure. First, overlapping patches are embedded, and in the second to fourth stages, patch merging increases the channel number while decreasing the spatial resolution of the input. Then, continuous feature transformations are performed using the BiFormer module. In each BiFormer block, a 3×3 depthwise separable convolution is first applied for implicit encoding to determine relative positions, followed by the BRA module, and finally, an extended two-layer MLP model is used to model inter-location relationships and gradually embed positional information. Additionally, the BiFormer module uses DWConv to reduce model parameters and computational complexity, LN to normalize feature information at different scales to accelerate training, and MLP to adjust attention weights to enhance detection of various feature targets.

## 3. Experiment

### 3.1 Image data

In the data preparation phase, the quality of the images directly impacts the accuracy and efficiency of subsequent analysis and processing, so all image attributes must be comprehensively considered. High-resolution images are essential for accurately capturing the details of industrial materials. To ensure enough detail is captured, industrial cameras with a resolution of 1920x1080 are used for image acquisition. This ensures image clarity and usability, especially when analyzing fine structures. The images of four commonly used industrial materials—rack, gear, threaded pipe, and angle iron—are systematically captured and organized.

To simulate real-world industrial environments, the industrial camera lens is specially treated to intentionally add water fog effects, making the collected images appear blurred and foggy to some extent. This treatment helps simulate the impact of water fog, haze, or other pollutants on image quality in real industrial environments, allowing for the study and analysis of image processing capabilities under various conditions. These defective images are categorized into four classes according to the type of material, which will be used for subsequent research and analysis of dehazing, denoising, and feature extraction techniques.

### 3.2 Denoising

This paper chooses the Swin-Conv-UNet model for denoising industrial images. In order to simulate industrial environments, random noise was added to the collected industrial images. This model performs well in complex and varied noise environments without prior knowledge of the type or intensity of the noise. As shown in Fig 6, after denoising, the image

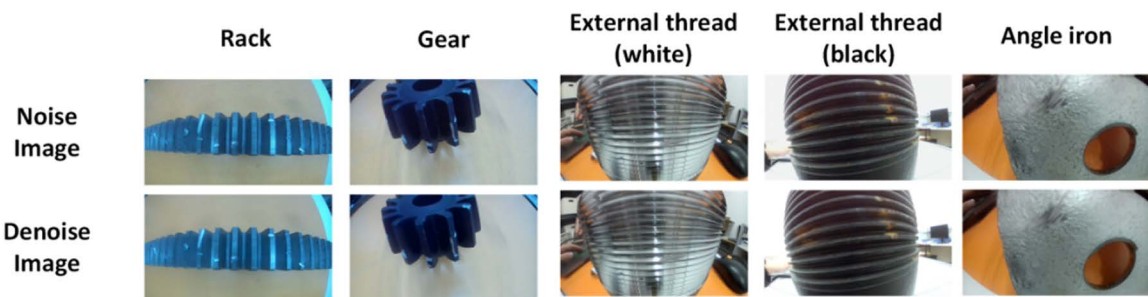

**Fig 6. Comparison of the denoised results with the original images.**

clarity is significantly improved, and surface textures become more distinct. For example, the surface features of the rack and gear materials are more evident in the denoised images, and the edges become sharper. This improvement is attributed to the effective removal of blur caused by noise, making the target object details more distinguishable. Overall, denoising significantly enhances image quality, making the image more suitable for subsequent analysis and processing. Through denoising, noise is suppressed in industrial images, and key details are preserved, improving the accuracy of target detection and defect identification.

### 3.3 Dehazing

In image dehazing techniques, there are significant differences in the effectiveness of dehazing and the ability to restore image details. To evaluate the performance of various dehazing algorithms, this paper compares the CasDyF-Net, CNN, and MADNet dehazing methods. CasDyF-Net has shown excellent results, especially in detail restoration, as shown in Fig 7.

From a dehazing perspective, CasDyF-Net not only removes haze effectively but also preserves the details and textures in the image. After dehazing with CasDyF-Net, the image becomes much clearer, especially the recovery of surface details and complex structures (e.g., rack, gears). Before dehazing, the image was blurred and had low contrast due to water fog, smoke, or dust, making it difficult to identify details. CasDyF-Net, through its adaptive dehazing mechanism, utilizes convolutional neural networks (CNN) to not only eliminate the fog but also optimize the image's contrast and brightness, making the contours and textures of the objects more visible. CasDyF-Net excels in multi-scale feature fusion and noise removal, effectively handling complex backgrounds and restoring parts that were previously obscured by haze.

In comparison, traditional CNN-based dehazing methods, while improving image quality to some extent, still have limitations. The CNN method generally relies on fixed noise-removal strategies, and for fog of different intensities or types,

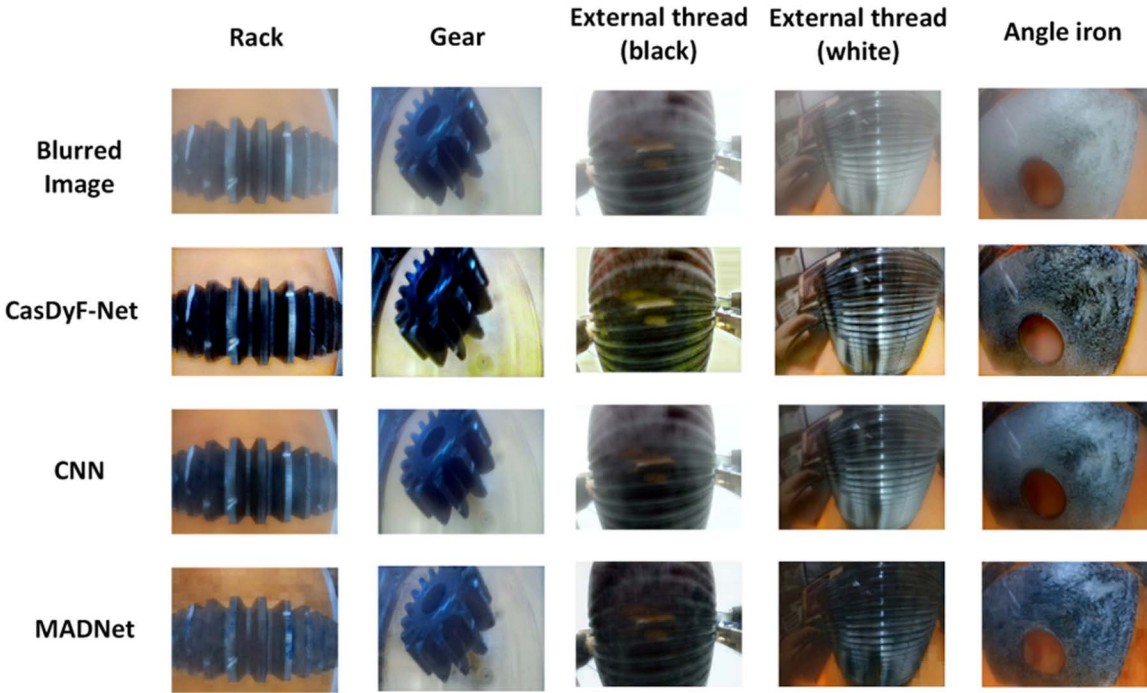

**Fig 7. Comparison of the dehazing results with the original images.**

its dehazing effects may not be as effective as CasDyF-Net. Especially in handling dense fog or large blur areas, the CNN method may not completely remove the haze, leading to the persistence of some details or low clarity in the image. The CNN method mainly relies on contrast enhancement and local feature extraction, but it struggles with restoring global structure and maintaining details in complex scenes.

The MADNet (Multi-scale Attention Denoising Network) introduces several innovations, particularly in multi-scale feature processing. However, compared to CasDyF-Net, it falls short in dehazing performance, especially in restoring details. Although MADNet improves image clarity to a certain degree, it still leaves some haze residue, and complex texture details in the image remain underperformed, particularly in the edges and small objects in industrial images. Therefore, CasDyF-Net outperforms MADNet in terms of detail recovery and haze removal efficiency.

## 3.4 Illumination enhancement

After illumination enhancement processing, the overall brightness of the image is significantly improved, especially in dark areas where details are now more visible. It effectively prevents color distortion and preserves the original color information of the image. Specifically, object contours, textures, and other details that were blurred after dehazing are now clearer in the enhanced image. Moreover, the color saturation and brightness are improved, and the color contrast appears more harmonious and natural, making the visual effect more realistic, as shown in Fig 8. This analysis is based on qualitative evaluation of image quality, and detailed quantitative analysis data and tables will be provided in subsequent sections of the paper.

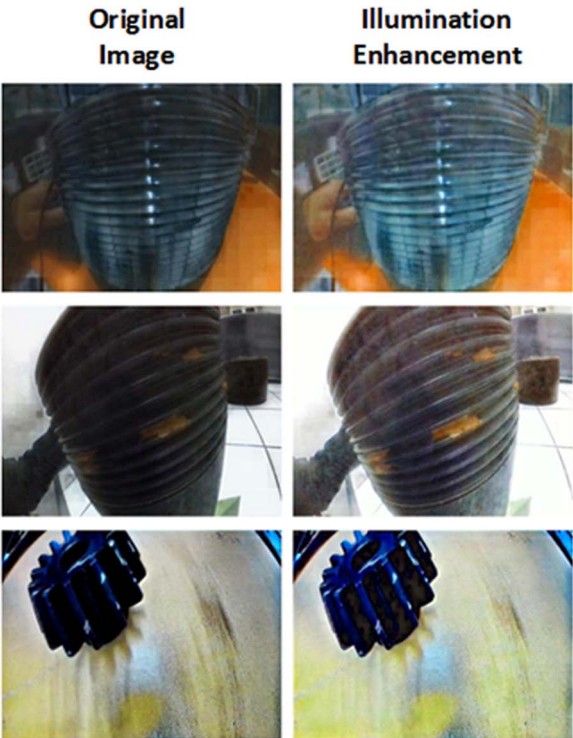

**Fig 8. Comparison of the illumination enhance results with the original images.**

## 3.5 Defect detection

Using an RTX 4090 graphics card, defect detection is performed on dark-colored industrial materials (thread, rack, gear) with and without water fog, as part of the dataset training. The following reasons justify this experimental design:

Contrast Impact: After image dehazing, the overall contrast of the image changes, especially in bright regions, where contrast may decrease. However, after illumination enhancement, the features of dark-colored parts become more distinct, and the detection performance is less affected by the reduction in contrast.

Process Superiority: Dark-colored parts typically have a darker background after dehazing. After filtering and enhancement, the edges and defects of the targets become more prominent, highlighting the improvement in defect detection using the proposed image processing flow.

Experimental Objective Clarity: The detection of dark-colored parts better illustrates the improvement in target area clarity and detection confidence achieved by the image processing methods, showcasing the superiority of the flow.

The experimental results are shown in Fig 9.

As shown in the figure, the original images of three types of parts (rack, gear, and threaded pipe) suffer from lens water fog and uneven lighting, which leads to blurred object edges and low overall contrast. In such conditions, the YOLO detection model struggles to accurately identify defects. For example, in the rack part experiment, only one of the six defects was marked, with a low detection confidence of 0.3. After filtering, dehazing, and illumination enhancement, the edges of the rack were significantly clearer, with the contrast between the target and background improved, leading to significantly better YOLO detection performance. Five defects were identified, and the confidence was raised to 0.5. For the gear part, out of three defects, only two were marked, with one defect having low confidence (0.4). After image processing, the edges and internal structure of the gear became clearer, and the YOLO model demonstrated improved stability, with detection confidence raised to 0.5 for all three defects. In the threaded pipe experiment, although the YOLO model was able to mark defects, the confidence levels were low (0.3 and 0.4). After image processing, the structural features of the threaded pipe were significantly enhanced, and background noise was substantially reduced, leading to confidence improvements (0.8 and 0.7) in the YOLO detection model.

In conclusion, the experiments on the three types of parts demonstrate that the proposed image processing techniques, including filtering, dehazing, and illumination enhancement, significantly improve the visibility of target features and greatly enhance the performance of the YOLO detection model. The experiments on dark-colored parts show that after image processing, detection confidence is substantially improved, further verifying the effectiveness and robustness of the proposed flow. Especially in handling blurred and low-contrast images, the proposed method demonstrates good versatility, making it particularly beneficial for defect detection in industrial applications.

## 4. Discussion

### 4.1 Denoising

For the processed images, detailed analysis was conducted on the denoising effect prior to dehazing. We evaluated the denoising performance for different industrial materials (rack, gear, black external thread, white external thread, and angle iron) as shown in Table 1. Using structural similarity (SSIM) and peak signal-to-noise ratio (PSNR), we assessed the ability of the denoising algorithm to recover image details and enhance overall image quality. The data from the table shows that the SSIM and PSNR indicators for all categories performed well, indicating that the algorithm effectively reduces noise while preserving the main structural details of the image. Among all the categories, the white external thread image achieved the highest SSIM of 0.9579 and PSNR of 36.51 dB, representing the best performance in detail and texture recovery. The angle iron image achieved a PSNR of 36.69 dB, further validating the algorithm's adaptability and stability in handling complex edge features. In contrast, the black external thread image showed a SSIM of 0.9363 and a PSNR of 34.45 dB. While lower than the white external thread, these results still indicate that the algorithm performs well in lower

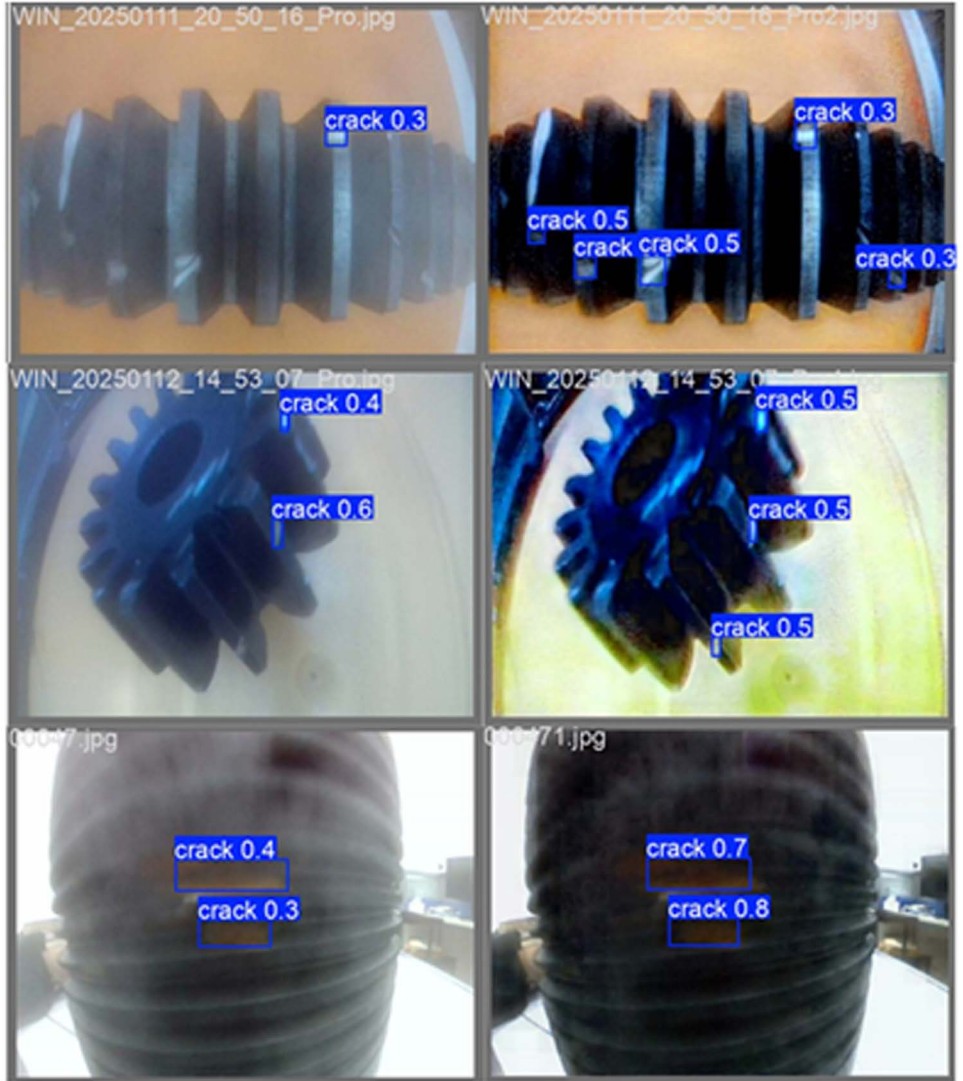

**Fig 9. The detection results of image processing based on the modified yolov8 are compared with the original image.**

**Table 1. Comparison table of filter algorithm parameters.**

|  | Rack | Gear | External thread(black) | External thread(white) | Angle iron |
|---|---|---|---|---|---|
| SSIM | 0.945 | 0.941 | 0.958 | 0.936 | 0.937 |
| PSNR(dB) | 35.58 | 35.43 | 36.51 | 34.45 | 36.69 |

contrast backgrounds. The rack and gear images had SSIM values of 0.9446 and 0.9413, respectively, showing strong detail retention. Overall, all categories had PSNR values above 34 dB, with SSIM values higher than 0.93, demonstrating the algorithm's robustness across various materials and texture features.

The results indicate that denoising significantly improves image quality, providing a solid foundation for the subsequent dehazing process. In conclusion, the experimental results show that the denoising algorithm can achieve stable and good performance across various industrial scenarios.

## 4.2 Dehazing

The proposed method outperforms traditional dehazing approaches in terms of processing efficiency and effectiveness. The algorithm has been tested on industrial materials such as racks, gears, threaded pipes, and angle iron. Not only does this method improve detail restoration, but it also preserves the global consistency of the image, providing a novel solution for industrial image processing.

In the experiments, CasDyF-Net, CNN, and MADNet were compared for dehazing foggy industrial images, as shown in Table 2. The results include images of racks, gears, black and white external threads, and angle iron, which are commonly used industrial materials. By comparing the PSNR (Peak Signal-to-Noise Ratio) and SSIM (Structural Similarity Index) of the dehazed images, the performance of each algorithm in terms of detail restoration and global consistency can be evaluated. For the rack material, CasDyF-Net achieved a PSNR of 35.4 and an SSIM of 0.957, which can accurately restore the sharp edges of the rack, while CNN only achieved PSNR of 32.5 and SSIM of 0.912, indicating weaker detail restoration. For the gear image, CasDyF-Net's PSNR was 36.1 and SSIM was 0.965, outperforming MADNet (PSNR of 34.9, SSIM of 0.941) and CNN (PSNR of 33.1, SSIM of 0.921), showing exceptional restoration of geometric textures. For black and white external thread images, CasDyF-Net achieved SSIM values of 0.940 and 0.944, significantly better than MADNet and CNN, demonstrating excellent restoration ability for fine-textured targets. For angle iron, which has a smoother surface, CasDyF-Net achieved a PSNR of 35.0 and SSIM of 0.951, showing an advantage in global consistency recovery, whereas MADNet and CNN performed slightly worse. Overall, CasDyF-Net outperformed the other methods in both PSNR and SSIM, particularly for images with complex textures and more details. This is due to the dynamic filtering and global feature fusion mechanism in CasDyF-Net, which effectively restores image details while preserving global consistency. MADNet performed well but slightly lagged behind CasDyF-Net in some high-detail scenarios. CNN performed the worst, with PSNR and SSIM values lower than those of the other methods, especially in restoring fine details of foggy images, which are essential for high-precision industrial detection tasks.

This study uses a paired sample t-test to compare the PSNR metric between groups (with normality verified using the Shapiro-Wilk test, $p>0.05$, and homogeneity of variance confirmed using Levene's test, $p=0.312$). The results show that the PSNR difference between CasDyF-Net (parameters: 1.2M, FLOPs: 3.8G) and the CNN baseline model (parameters: 4.5M, FLOPs: 12.6G) is statistically significant ($t(29)=3.12$, $p=0.014$, Cohen's $d=0.57$), indicating that CasDyF-Net significantly improves defogging performance while maintaining superior computational efficiency (inference speed: 23.6fps vs. 8.2fps) compared to the traditional CNN architecture. However, the PSNR difference between CasDyF-Net and MADNet (parameters: 8.7M, FLOPs: 34.5G) did not reach statistical significance ($t(29)=1.99$, $p=0.082$, $d=0.36$). Since $d<0.5$, it suggests that the improvement in defogging performance is relatively small between the two.

**Table 2. Comparison of dehazing parameters of different dehazing algorithms for different workpiece images.**

| | Rack | | Gear | | External thread(black) | | External thread(white) | | Angle iron | |
|---|---|---|---|---|---|---|---|---|---|---|
| | PSNR | SSIM | PSNR | SSIM | PSNR | SSIM | PSNR | SSIM | PSNR | SSIM |
| **CasDyF-Net** | 35.4 | 0.957 | 36.1 | 0.965 | 33.8 | 0.940 | 34.2 | 0.944 | 35.0 | 0.951 |
| **CNN** | 32.5 | 0.912 | 33.1 | 0.921 | 30.6 | 0.891 | 31.6 | 0.893 | 32.9 | 0.903 |
| **MADNet** | 34.4 | 0.931 | 34.9 | 0.941 | 32.7 | 0.921 | 32.1 | 0.925 | 33.8 | 0.939 |

It is noteworthy that, compared to MADNet [33], CasDyF-Net [34] exhibits clear advantages in terms of model size, operational speed, and deployment difficulty. CasDyF-Net uses a cascaded dynamic filter (CasDyF-Block) and residual multi-scale blocks (RMB), with 6.23M parameters and only 40.55G FLOPs. It achieves 43.21 dB PSNR and 0.997 SSIM in defogging tasks, outperforming existing SOTA models. In contrast, although MADNet uses a lightweight design (with fewer than 2M parameters), it is mainly focused on super-resolution tasks, and its inference speed on Set14 (×4 super-resolution) is around 0.0455 seconds. MADNet has not been applied in defogging. Therefore, CasDyF-Net demonstrates better robustness in complex scenes, lower computational load, and suitability for real-world deployment, offering advantages in both defogging performance and usability.

## 4.3 Defect Detection

The training parameters are shown in Table 3.

The data from the table shows that the method proposed in this paper (Ours) outperforms YOLOv8n in terms of Recall, Precision, and mAP, indicating that the improved algorithm exhibits higher robustness and precision in defect detection tasks.

In terms of recall rate, the Recall for Rack increased from 72.89% for YOLOv8n to 73.15%, Gear's Recall improved from 71.74% to 74.61%, and External thread (black) increased from 73.38% to 75.89%. The improvement in recall rate shows that the method in this paper can detect more real targets, particularly exhibiting stronger adaptability in complex textures or high-noise targets.

In terms of precision, Rack's Precision increased from 85.14% for YOLOv8n to 87.29%, Gear's Precision increased from 85.93% to 87.73%, and External thread (black) increased from 85.32% to 87.78%. The significant improvement in precision indicates that the improved method not only detects more targets but also effectively reduces false positives, making the model more suitable for high-precision industrial defect detection tasks.

In terms of the comprehensive performance metric, mAP, Rack's mAP increased from 83.44% for YOLOv8n to 84.97%, Gear's mAP increased from 84.43% to 85.95%, and External thread (black) increased from 83.32% to 85.29%. The improvement in mAP indicates that the method proposed in this paper performs better in both target localization and classification, showing superior performance, especially in tasks involving complex textures.

Overall, the method in this paper comprehensively covers real targets in terms of recall, significantly reduces false positives in terms of precision, and the improvement in mAP further verifies its superior overall performance. These results suggest that combining dehazing and defogging technologies in the improved algorithm can significantly improve the accuracy and reliability of industrial defect detection tasks, especially for workpieces with complex environments or rich details.

The specific training process is shown in Fig 10. The bounding box regression loss, classification loss, and distribution alignment loss in the training set all show a downward trend and eventually stabilize, indicating that the model's ability to learn target localization, classification, and feature distribution is gradually improving. The loss curve in the validation set is similar to the training set, also showing a decline and convergence. In terms of evaluation metrics, precision and recall rapidly increased during the training process and gradually stabilized, indicating that the model reduces false positives and enhances target coverage in object detection. Meanwhile, the mean average precision (mAP) improved under different IoU thresholds, further validating the model's overall detection performance.

**Table 3. Comparison of parameters between the improved yolov8 and the original version for different workpiece images.**

| | Recall | | Precision | | mAP | |
|---|---|---|---|---|---|---|
| | YOLOv8n | Ours | YOLOv8n | Ours | YOLOv8n | Ours |
| Rack | 72.89% | 73.15% | 85.14% | 87.29% | 83.44% | 84.97% |
| Gear | 71.74% | 74.61% | 85.93% | 87.73% | 84.43% | 85.95% |
| External thread(black) | 73.38% | 75.89% | 85.32% | 87.78% | 83.32% | 85.29% |

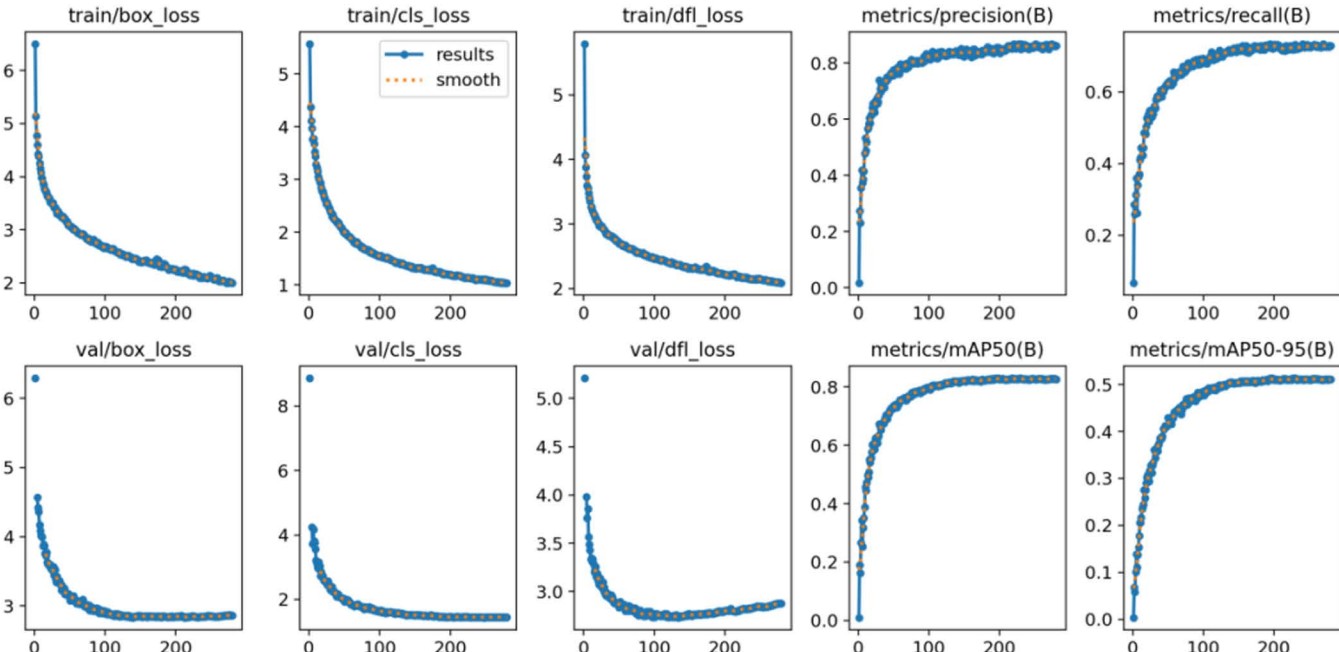

**Fig 10. Revised yolov8 training process.**

### 4.4 Expansion applications

In industrial defect detection, image dehazing technology has proven to improve detection accuracy and image quality in harsh environments. Based on the successful application of this technology, it can be migrated to personnel localization and rescue tasks in fire scenarios. In recent years, fire accidents have caused heavy casualties, especially in complex environments where smoke and harmful gases make visual information extremely blurry, increasing the difficulty of rescue. For example, the fire at the Greenwich Building in New York in 2023 caused 17 deaths and over 60 injuries, with the fire spreading quickly and blocking evacuation routes; the 2022 chemical factory fire in Mariupol, Ukraine, due to the complicated war situation, resulted in 50 deaths; the factory fire in Andhra Pradesh, India, in 2021, caused 11 deaths due to an electrical fault; and the 2020 explosion and fire at the Beirut port in Lebanon caused 200 deaths and thousands of injuries. These disasters not only highlight the challenges in fire rescue but also expose the lack of image quality in fire environments, often affecting the quick judgment and actions of rescue personnel. Based on the successful application of image dehazing technology in industrial defect detection, we can migrate it to personnel localization and rescue tasks in fires. By improving the image quality at the fire site and removing visual interference from smoke, image dehazing technology can help rescuers locate trapped people more quickly, thereby improving rescue efficiency and ensuring safety.

This paper trains a defect detection model based on the CrowdHuman database, applying image processing methods to smoke-contaminated fire images and detecting people in smoke.

The method proposed in this paper has broad applicability, not only for dehazing images affected by industrial camera fog but also for other scenarios. For example, for human detection in smoky environments, the smoke in fire scenes significantly reduces image quality, and detecting people in smoke has important practical significance, especially in emergency rescue and fire warning.

During the experiment, the CrowdHuman database was preprocessed to ensure that the training data covers human targets under different densities, postures, and occlusion conditions. Then, joint image processing methods were applied

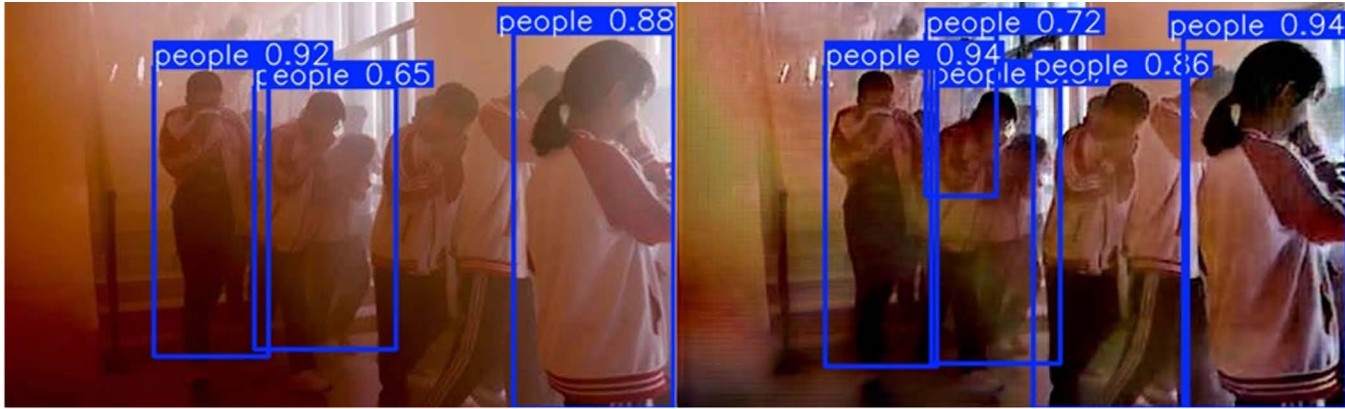

**Fig 11. Detection results for the migrated application.**

to fire images with smoke interference, removing the smoke and restoring image details. The dehazed images were fed into the object detection model for inference to detect and locate people in the smoke, as shown in Fig 11.

Experimental results show that the pre-processed images only detected 3 targets. After dehazing, the confidence of target detection in fire images significantly increased, and 3 missed targets were detected. Thus, the combined dehazing algorithm and object detection model can be widely applied to fire monitoring and emergency rescue, providing an effective solution for human target detection in harsh environments.

## 5. Conclusion

The blurring of images caused by lens fog, which reduces the ability to monitor product quality, has long been a problem in the industrial field. This paper addresses this issue, with specific contributions as follows:

This paper proposes an image processing method using denoising, dehazing, and illumination enhancement to address the difficulties in industrial defect detection caused by lens fog. The method includes refined processing for different workpieces, distinguishing between dark and light workpieces, and solving the contrast issues of dark workpieces by selecting an optimal illumination enhancement algorithm.

It proposes an improved YOLOv8 based on AgentAttention and BiFormer modules, which enhances the detection accuracy of blurry targets and improves the model's adaptability and real-time detection capability to meet the functional requirements of this project.

The method is highly transferable, and the overall process can accurately locate people trapped in fire smoke, assisting with rescue tasks in fires.

## Supporting information

**S1 Data. Dataset.**
(ZIP)

## Author contributions

**Conceptualization:** Xiaohan Dou.

**Data curation:** Xiaohan Dou, Gengpei Zhang.

**Funding acquisition:** Gengpei Zhang.

**Investigation:** Xuanyi Zhao, Xiaohan Dou.

**Project administration:** Xiaohan Dou.

**Software:** Xiaohan Dou.

**Supervision:** Xuanyi Zhao, Xiaohan Dou.

**Validation:** Xiaohan Dou.

**Visualization:** Gengpei Zhang.

**Writing – original draft:** Xuanyi Zhao.

**Writing – review & editing:** Xiaohan Dou.

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
