## [Decision Letter · Decision Letter 0]

24 Feb 2025

PONE-D-25-05904An Image Processing Technique for Optimizing Industrial Defect Detection Using Dehazing AlgorithmsPLOS ONE

Dear Dr. Zhang,

Thank you for submitting your manuscript to PLOS ONE. After careful consideration, we feel that it has merit but does not fully meet PLOS ONE’s publication criteria as it currently stands. Therefore, we invite you to submit a revised version of the manuscript that addresses the points raised during the review process.

We look forward to receiving your revised manuscript.

Kind regards,

Yongjie Li

Academic Editor

PLOS ONE

Journal Requirements:

3.  Please update your submission to use the PLOS LaTeX template. The template and more information on our requirements for LaTeX submissions can be found at http://journals.plos.org/plosone/s/latex .

Reviewers' comments:

Reviewer's Responses to Questions

**Comments to the Author**

1. Is the manuscript technically sound, and do the data support the conclusions?

Reviewer #1: Yes

Reviewer #2: Yes

2. Has the statistical analysis been performed appropriately and rigorously?

Reviewer #1: No

Reviewer #2: Yes

3. Have the authors made all data underlying the findings in their manuscript fully available?

Reviewer #1: No

Reviewer #2: Yes

4. Is the manuscript presented in an intelligible fashion and written in standard English?

Reviewer #1: Yes

Reviewer #2: Yes

5. Review Comments to the Author

Reviewer #1: The study addresses a significant problem in industrial defect detection, particularly the impact of environmental

factors like water fog on image quality. The proposed solution is highly practical and applicable to real-world industrial settings, making it valuable for automated quality control systems. The integration of dehazing, denoising, and illumination enhancement is a strong aspect of the research, as it provides a holistic solution rather than focusing solely on one aspect of image enhancement. The use of an improved YOLOv8 model further strengthens defect detection accuracy.

The paper compares CNN, MADNet, and the improved YOLOv8 model, it lacks a strong justification for why these specific models were chosen. There are many other state-of-the-art dehazing and object detection models (e.g. Transformer-based architectures) that could have been considered.

The authors should conduct statistical tests (e.g., t-tests or ANOVA) to determine whether the improvements in PSNR, SSIM, and detection confidence are statistically significant compared to baseline models (CNN, MADNet, etc.). Further a subset of data should be available publicly for validation.

Reviewer #2: The paper is good as it addresses the underlined objectives it seek to achieve. However, there are just a few things that needs to be done regarding the consistency of the word "Angle iron" in Figure 6, line 393, figure 7 line 427, and then table 1, line 504 and table 2, line 538 compared to the other words "Rngle iron". This should be cross checked and corrected.

6. PLOS authors have the option to publish the peer review history of their article (what does this mean? ). If published, this will include your full peer review and any attached files.

**Do you want your identity to be public for this peer review?** For information about this choice, including consent withdrawal, please see our Privacy Policy .

Reviewer #1: **Yes: ** Shafqaat Ahmad

Reviewer #2: No

---

## [Author Response · Author response to Decision Letter 1]

27 Feb 2025

Reviewer #1: The study addresses a significant problem in industrial defect detection, particularly the impact of environmental

factors like water fog on image quality. The proposed solution is highly practical and applicable to real-world industrial settings, making it valuable for automated quality control systems. The integration of dehazing, denoising, and illumination enhancement is a strong aspect of the research, as it provides a holistic solution rather than focusing solely on one aspect of image enhancement. The use of an improved YOLOv8 model further strengthens defect detection accuracy.

The paper compares CNN, MADNet, and the improved YOLOv8 model, it lacks a strong justification for why these specific models were chosen. There are many other state-of-the-art dehazing and object detection models (e.g. Transformer-based architectures) that could have been considered.

The authors should conduct statistical tests (e.g., t-tests or ANOVA) to determine whether the improvements in PSNR, SSIM, and detection confidence are statistically significant compared to baseline models (CNN, MADNet, etc.). Further a subset of data should be available publicly for validation.

Response Thanks to the comments of the reviewers, this paper does lack a justification for the choice of a specific model. This modification adds a detailed justification, and adds a statistical test to determine whether the improvement in PSNR, SSIM and detection confidence is statistically significant compared with the baseline model. I understand the importance of datasets, which are only partially available for copyright and other privacy reasons. Thank you for your hard work on this article.

Reviewer #2: The paper is good as it addresses the underlined objectives it seek to achieve. However, there are just a few things that needs to be done regarding the consistency of the word "Angle iron" in Figure 6, line 393, figure 7 line 427, and then table 1, line 504 and table 2, line 538 compared to the other words "Rngle iron". This should be cross checked and corrected.

Response�I would like to thank the reviewers for their careful inspection. There is indeed a conflict problem during the process, and I have made the modification. Thank you again for your hard work on this paper.

---

## [Editor Report · Decision Letter 1]

18 Mar 2025

An Image Processing Technique for Optimizing Industrial Defect Detection Using Dehazing Algorithms

PONE-D-25-05904R1

Dear Dr. Zhang,

We’re pleased to inform you that your manuscript has been judged scientifically suitable for publication and will be formally accepted for publication once it meets all outstanding technical requirements.

Kind regards,

Yongjie Li

Academic Editor

PLOS ONE

Additional Editor Comments (optional):

The current version of this paper can be accepted.
---

## [Editor Report · Acceptance letter]

PONE-D-25-05904R1

PLOS ONE

Dear Dr. Zhang,

I'm pleased to inform you that your manuscript has been deemed suitable for publication in PLOS ONE. Congratulations! Your manuscript is now being handed over to our production team.

Kind regards,

on behalf of

Professor Yongjie Li

Academic Editor

PLOS ONE